

# Development of high-resolution Thermosphere-Ionosphere Electrodynamics General Circulation Model (TIE-GCM) using Ring Average technique

**Tong Dang**[1,2,3,5]**, Binzheng Zhang**[4,5]**, Jiuhou Lei**[1,2,3,*]**, Wenbin Wang**[5]**, Alan Burns**[5]**, Han-li Liu**[5]**, Kevin Pham**[5]**, Kareem A. Sorathia**[6]

[1]CAS Key Laboratory of Geospace Environment, School of Earth and Space Sciences, University of Science and Technology of China, Hefei, China

[2]Mengcheng National Geophysical Observatory, University of Science and Technology of China, Hefei, China

[3]CAS Center for Excellence in Comparative Planetology, Hefei, China

[4]Department of Earth Sciences, the University of Hong Kong, Pokfulam, Hong Kong SAR, China

[5]High Altitude Observatory, National Center for Atmospheric Research, Boulder, CO, USA

[6]Applied Physics Laboratory, Johns Hopkins University, Laurel, MD, USA

[*]Corresponding author: Jiuhou Lei, leijh@ustc.edu.cn

**Abstract.** When solving hydrodynamic equations in spherical/cylindrical geometry using explicit finite difference schemes, a major difficulty is that the time step is greatly restricted by the clustering of azimuthal cells near the pole due to the Courant-Friedrichs-Lewy condition. This paper adapts the azimuthal averaging-reconstruction (Ring Average) technique to finite difference schemes in order to mitigate the time step constraint in spherical/cylindrical coordinates. The finite-difference Ring Average technique averages physical quantities based on an effective grid and then reconstructs the solution back to the original grid in a piece-wise, monotonic way. The algorithm is implemented in a community upper atmospheric model Thermosphere-Ionosphere Electrodynamics General Circulation Model (TIE-GCM), with horizontal resolution up to $0.625° \times 0.625°$ in the geographic longitude-latitude coordinates, which enables the capability of resolving critical mesoscale structures within the TIE-GCM. Numerical experiments have shown that the Ring Average technique introduces minimal artifacts in the polar region of the GCM solutions, which is a significant improvement compared to the commonly used low-pass filtering techniques such as the fast Fourier transform method. Since the finite-difference adaption of the Ring Average technique is a post-solver type algorithm, which requires no changes to the original computational grid and numerical algorithms, it has also been implemented in much more complicated models with extended physical/chemical modules such as the coupled Magnetosphere Ionosphere Thermosphere (CMIT) model and the Whole Atmosphere Community Climate Model with thermosphere and ionosphere eXtension (WACCM-X). The implementation of the Ring Average techniques in both models enables CMIT and WACCM-X to perform global simulations with a much higher resolution than that used in the community versions. The new technique is not only a significant improvement in space weather modeling capability, but can also be adapted to more general finite difference solvers for hyperbolic equations in spherical/polar geometries.

**Key Points:**

- Ring Average technique is adapted to solve the issue of clustered grid cells in polar/spherical coordinate with finite difference method.

- The Ring Average technique is applied to develop a $0.625° \times 0.625°$ high-resolution TIE-GCM and more complicated geoscientific models.

- The high-resolution TIE-GCM shows good capability in resolving mesoscale structures in the I-T system.

**Keywords:** Finite difference method, Spherical geometry, Ionosphere-thermosphere system, General circulation model, CFL condition.



# 1 Introduction

Mesoscale structures with typical horizontal size of $100{\sim}500$ km, have gained more and more attention in the research of the dynamics of the upper atmospheric system. A number of studies have been carried out to investigate these structures, including the formation and evolution of polar cap patches and tongues of ionization [Basu et al., 1995; Foster et al., 2005; Zhang et al., 2013], dynamics of ionospheric irregularities [Makela and Otsuka, 2012; Sun et al., 2015], variations of polar thermospheric density anomaly [Crowley et al., 2010; Lühr et al., 2004], and the space weather effects of mesoscale electric field variability [Codrescu et al., 1995; Matsuo and Richmond, 2008; Zhu et al., 2018; Lotko and Zhang, 2018]. These dynamic mesoscale structures have shown critical importance in both understanding the physics of the solar-terrestrial system and space weather predictions, which challenges the resolution and accuracy of numerical models of the upper atmospheric system in resolving these important mesoscale signatures.

Spherical or cylindrical coordinates are commonly used in solving geophysical problems, including the modeling of the upper atmospheric systems. As a workhorse for space weather research, a number of global circulation models (GCMs) for the coupled ionosphere-thermosphere (I-T) system have been developed based on spherical coordinates using finite difference schemes [e.g., Richmond et al., 1992; Fuller-Rowell et al., 1996; Ridley et al., 2006; Ren et al., 2009]. However, restricted by the longitudinal grid resolution, current horizontal resolutions used in I-T GCMs are still insufficient for fully resolving mesoscale atmospheric structures, which are either marginal or sub-grid. The latest released version of the community code, Thermosphere Ionosphere Electrodynamic General Circulation Model (TIE-GCM), has a longitude-latitude resolution of $2.5° \times 2.5°$; the Coupled Thermosphere/Ionosphere Plasmasphere (CTIP) Model has a latitude resolution of $2°$ and a longitude resolution of $18°$; the most recent version of the Global Ionosphere Thermosphere Model (GITM) has a flexible grid with latitudinal resolution up to $0.3125°$ but the typical longitudinal resolution remains $2.5°$ due to severe time step restrictions for global-scale calculations.

The major difficulty in increasing longitudinal resolution in spherical geometry based GCMs is that the explicit time stepping is constrained by the clustering azimuthal cells near the pole due to the Courant-Friedrichs–Lewy (CFL) condition [Courant et al., 1928]. A number of attempts have been proposed to address this coordinate singularity issue ([e.g., Purser, 1988; Bouaoudia and Marcus, 1991; Williamson et al., 1992; Takacs, 1999; Fukagata and Kasagi, 2002; Prusa, 2018]). To use a time step that is larger than the global minimum requirement from the CFL conditions, one common method used in a spherical GCM is to employ a low-pass Fourier filter at polar latitudes, which removes non-physical, high-frequency zonal





waves generated due to numerical instability caused by the local violation of the CFL conditions [e.g., Ska-

marock et al., 2008]. Although the Fourier filter can maintain the computational stability and permit a much

larger temporal step, the applicability of the Fast Fourier Transform (FFT) filter method is problem depen-

dent, which also bring barriers in moving models forward to finer spatial resolutions. Moreover, the linear

filtering of zonal components generated through a non-physical time step may decrease the accuracy of the

model calculations near the polar region, which affects physical conservations of e.g. mass, momentum and

energy that are essential for long-term behavior of the GCM [Williamson and Browning, 1973].

Recently, Zhang et al. [2019] developed a new technique named the "Ring Average" method for hy-

perbolic equations to mitigate the CFL restrictions in spherical polar geometry, on the basis of the method

originally proposed in the Lyon-Fedder-Mobarry (LFM) MHD simulations [Lyon et al., 2004]. The method

is a "post-solver" type algorithm applied after solving all the physical quantities in the original spherical

polar coordinates, thus no modification to either the numerical solver or the computational grid is required

when applying the Ring Average. Test simulation results have shown the effectiveness of the Ring Average

algorithm in increasing the time step by a factor of 100 while maintaining the fidelity of the solutions. The

original Ring Average technique was developed for solving hyperbolic equations in spherical/polar geom-

etry based on finite volume schemes, which redistributes the solution azimuthally through a conservative

averaging-reconstruction algorithm. The finite-volume version of the Ring Average technique not only re-

leases the time step constraint in spherical geometry, but also keeps the conservative nature of finite-volume

schemes to machine precision. In this paper, we adapt the Ring Average technique to finite difference

schemes for solving hyperbolic equations. Defined on an effective reduced polar grid, the finite-difference

adaption of the Ring Average technique also conducts a "post-solver" step of averaging-reconstruction in

each azimuthal ring to maintain the numerical stability and relax the severe computational time step con-

straint. To demonstrate the effectiveness of the finite-difference version of the Ring Average technique,

we use solutions from both linear advection equations and the TIE-GCM as test beds. The Ring Average

algorithm enables the use of high-resolution TIE-GCM such as $0.625° \times 0.625°$ in longitude and latitude

with reasonable time steps and minimal numerical artifacts. Further applications of the technique on cou-

pled Magnetosphere Ionosphere Thermosphere (CMIT) and Whole Atmosphere Community Climate Model

with thermosphere and ionosphere extension (WACCM-X) are also addressed.

This paper is organized as follows: In Section 2, we describe the details of the model and the Ring

Average technique to solve the clustering of polar grid cells problem. A hydrodynamic convection experi-





ment with the Ring Average technique has also been conducted to test the capability of the method. Section

3 shows the preliminary results of the high-resolution TIE-GCM with the Ring Average technique imple-

mented as well as the further applications of the technique. The findings of this work are summarized in

Section 4.

## 2 Methodology

### 2.1 Ring Average in the Finite-Difference Form

An example of the standard polar grids with a horizontal resolution of $2.5° \times 2.5°$ (longitude×latitude)

in the TIE-GCM is shown in Figure 1. It is evident that in Figure 1a, the azimuthal (longitudinal) com-

putational nodes in the standard polar grid are significantly clustered near the pole even with 144 cells in

the azimuthal direction, resulting in very "thin" cells with small azimuthal extensions which restricts the

explicit time step for the advection equations. This azimuthal clustering becomes even worse when grid

resolution increases, namely the time step drops to 1/4 while the grid resolution doubles, corresponding

to at least 32 times increases in computational resource, which becomes expensive especially for global

simulations with high spatial resolutions in order to resolve mesoscale structures.

The finite difference adaption of the Ring Average algorithm is based on a similar averaging-reconstruction

process over a pre-defined, "effective" azimuthal grid as used in the finite-volume version of the algorithm.

Figure 1b shows an example of "effective" polar grid for applying the finite-difference Ring Average tech-

nique. In the polar grid shown in Figure 1b, since the reconstructed solution is monotonic within each

effective computational cell, a much larger time step is allowed compared to the original grid shown in

Figure 1a. As shown in Figure 1b, the effective longitudinal grid resolutions have been reduced and are

less clustered towards the pole. For the most inside (highest latitude) grids, the 144 azimuthal cells (Figure

1a) have been grouped to 9 effective cells (chunks), with 16 original cells in each chunk. Moving away

from the pole, more chunks are employed. As an example, the numbers of chunks from inside to outside

shown in Figure 1b in the effective grid are 9, 9, 18, 18, 36, 36, 72, 72, 72, and 72, respectively, allowing

a relative smooth transition in the size of the cells going radially outward. Note that the choice of the num-

ber of chunks in each ring is non-unique. Numerical tests with the Finite-Volume solvers have shown that

the computational solution, under both smooth and discontinuous flow conditions, is insensitive to small

changes in the chunk configuration [Zhang et al., 2019].





(a) Original Polar Grid        (b) Effective Grid

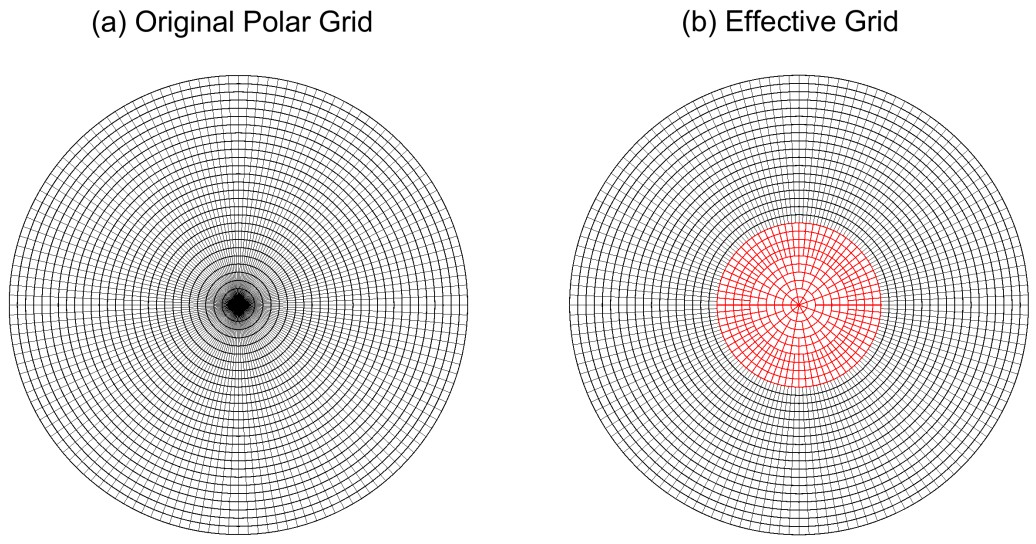

**Fig 1** The (a) original and (b) effective $2.5° \times 2.5°$ TIE-GCM polar longitude-latitude grid.

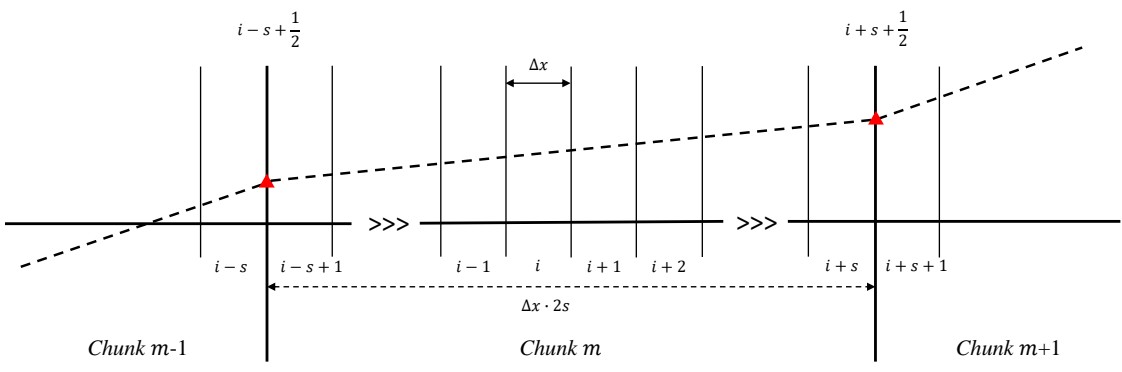

**Fig 2** Schematic of grid cells within effective chunks

We use the following example of solving the linear advection equation to illustrate the averaging-

reconstruction process within each chunk. Consider the following linear advection equation of an incom-

pressible fluid in the azimuthal direction as an example:

$$\frac{\partial \rho}{\partial t} + v\frac{\partial \rho}{\partial x} = 0, \tag{1}$$



where $v$ is the advection velocity, $\rho$ is the density profile, and $x$ is the azimuthal dimension ($x \in [0\ 2\pi]$) along one ring. Assuming the $x$ direction is uniformly discretized into $N_{total}$ computational cells with $\Delta x = \frac{2\pi}{N_{total}}$, a central difference Euler form of Equation (1) for density $\rho$ in cell $k$ between time $n$ and $n+1$ is written as

$$\frac{1}{\Delta t}\left(\rho_k^{n+1} - \rho_k^n\right) = -\frac{v}{2\Delta x}\left(\rho_{k+1}^n - \rho_{k-1}^n\right), \tag{2}$$

where $k$ denotes the index of an individual "thin" cell in the original azimuthal grid. $\Delta t$ is the time step regulated by the CFL condition. Without ring average type of treatment, the time step $\Delta t$ is restricted by the fact that "thin" azimuthal cells cluster near the pole. The ring average technique takes the average solution within a chunk $m$ that contains $2s$ cells in the original grid as shown in Figure 2. Summing over the finite-difference form of Equation (1) within chunk $m$ gives:

$$\sum_{k=i-s+1}^{i+s}\frac{1}{\Delta t}\left(\rho_k^{n+1} - \rho_k^n\right) = -\frac{v}{2\Delta x}\sum_{k=i-s+1}^{i+s}\left(\rho_{k+1}^n - \rho_{k-1}^n\right). \tag{3}$$

Then summing over the $k$ indices within chunk $m$, Equation (3) becomes:

$$\frac{1}{\Delta t}\left(\sum_{k=i-s+1}^{i+s}\rho_k^{n+1} - \sum_{k=i-s+1}^{i+s}\rho_k^n\right) = -\frac{v}{2\Delta x}\sum_{k=i-s+1}^{i+s}\left[\left(\rho_{k+1}^n - \rho_k^n\right) + \left(\rho_k^n - \rho_{k-1}^n\right)\right] \tag{4}$$

$$= -\frac{v}{2\Delta x}\left[\left(\rho_{i+s+1}^n - \rho_{i-s+1}^n\right) + \left(\rho_{i+s}^n - \rho_{i-s}^n\right)\right] \tag{5}$$

$$= -\frac{v}{\Delta x}\left(\frac{\rho_{i+s+1}^n + \rho_{i+s}^n}{2} - \frac{\rho_{i-s+1}^n + \rho_{i-s}^n}{2}\right) \tag{6}$$

$$= -\frac{v}{\Delta x}\left(\rho_{i+s+\frac{1}{2}}^n - \rho_{i-s+\frac{1}{2}}^n\right), \tag{7}$$

where $\rho_{i-s+\frac{1}{2}}^n$ and $\rho_{i+s+\frac{1}{2}}^n$ are the left- and right-values on the boundary of chunk $m$, as indicated by the red triangles in Figure 2. The LHS of Equation (7) is basically the time rate of the change in terms of the chunk density $\varrho_m$:

$$\frac{1}{\Delta t}\left(\varrho_m^{n+1} - \varrho_m^n\right), \tag{8}$$

where $\varrho_m^{n+1} = \sum_{k=i-s+1}^{i+s}\rho_k^{n+1}$ and $\varrho_m^n = \sum_{k=i-s+1}^{i+s}\rho_k^n$. If assuming smoothness of the solution which applies to typical upper atmospheric flow conditions, and using a piece-wise linear reconstruction for the





two interface values at time level $n$:

$$\rho^n_{i+s+\frac{1}{2}} = \frac{\varrho_{m+1} + \varrho_m}{2} \tag{9}$$

$$\rho^n_{i-s+\frac{1}{2}} = \frac{\varrho_m + \varrho_{m-1}}{2}, \tag{10}$$

the RHS of Equation (7) is in the form of a central difference approximation of the spatial derivative $\frac{\partial \varrho}{\partial x}$ in chunk $m$:

$$-\frac{v}{\Delta x} \frac{\varrho^n_{m+1} - \varrho^n_{m-1}}{2}. \tag{11}$$

Equating Equation (8) and Equation (11) and considering the fact that the $\Delta X$ in computing the chunk derivative is actually $2s\Delta x$, we obtain:

$$\frac{1}{\Delta T}\left(\varrho^{n+1}_m - \varrho^n_m\right) = -v\frac{\varrho^n_{m+1} - \varrho^n_{m-1}}{2\Delta X}, \tag{12}$$

where $\Delta T = 2s\Delta t$. Equation (12) is in the same numerical differential form of the advection equation in terms of the chunk density $\varrho$ in the effective grid, within the same order of finite difference approximation:

$$\frac{\partial \varrho}{\partial t} = -v\frac{\partial \varrho}{\partial x} + \mathcal{O}\left(\Delta X^2\right). \tag{13}$$

Equation (12) also suggests that in principle the ring average method is capable of using a time step that is approximately $2s$ times larger than the original $\Delta t$ restricted by the "thin" cells (assuming the CFL condition is dominated by the azimuthal direction in the innermost ring). Note that the above derivation of the finite-difference version of the Ring Average algorithm is independent of the numerical schemes solving the linear advection equation (1). Thus, the Ring Average algorithm requires no modifications to the existing hydrodynamic equations solved by GCMs. On the other hand, since the Ring Average algorithm is applied after all the variables are solved on the original spherical grid, it requires no changes to the existing computational grid.

In the reconstruction step, the above algorithm uses the piecewise linear method (PLM) to reconstruct solutions within each chunk for the next time step of the GCM calculations, resulting in a $2^{nd}$-order accuracy. To achieve higher accuracy in the reconstruction step, a piecewise parabolic reconstruction method (PPM) [Colella and Woodward, 1984] may be used in the algorithm, which provides a $4^{th}$-order accuracy for the





reconstruction step. In the following section when applying the Ring Average algorithm in a GCM, we

use both PPM and PLM for different variables. The criteria using PPM or PLM here depends on their

spatial gradient from the fluid calculations. For variables which have relatively greater spatial gradient, we

use PPM method to reach a high accuracy and maintain the stability, otherwise the PLM is used for the

calculations.

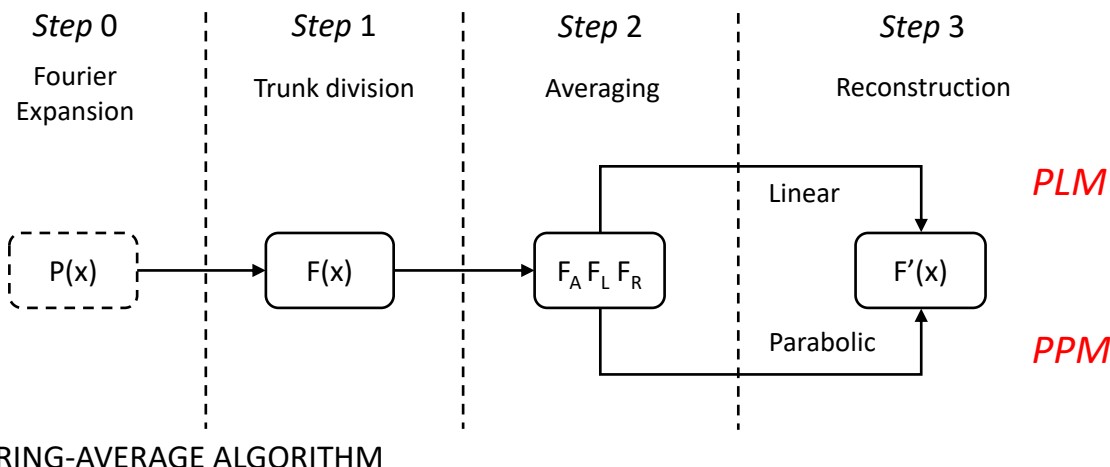

**Fig 3** The Ring Average Algorithm with both PLM and PPM methods

The algorithm shown in Figure 3 illustrates the steps of applying the Ring Average technique using either

PLM or PPM. The steps consist of chunk division, chunk averaging and reconstruction. The averaging-

reconstruction process (Ring Average) in this study is similar to Zhang et al. [2019], with modifications on

the reconstruction method (PPM or PLM) adapted to finite difference schemes. The detailed procedures of

Ring Average technique in this study are described as follows:

*For variables using the PLM reconstruction*

*Step 1.* Divide the azimuthal grid cells into chunks and pull data into the chunks.

*Step 2.* Calculate the average value $F_A$, left interface value $F_L$, and right interface value $F_R$ at chunk

$m$ ($m$ is the index of the chunk number in an azimuthal ring). $F_L$ and $F_R$ are the interface values in each





chunk and determined by the following parabola functions:

$$F_L = (-F_{m-2} + 7F_{m-1} + 7F_m - F_{m+1})/12 \tag{14}$$

$$F_R = (-F_{m-1} + 7F_m + 7F_{m+1} - F_{m+2})/12, \tag{15}$$

where $F_{m-2}$, $F_{m-1}$, $F_m$, $F_{m+1}$, and $F_{m+2}$ are the average values $F_A$ at chunks with index of $m-2$, $m-1$, $m$, $m+1$, $m+2$, respectively.

*Step 3*. Reconstruct the variables by interpolating the average data linearly in each chunk:

$$F_k = (1 - \frac{k}{N})F_L + \frac{k}{N}F_R, \tag{16}$$

where $N$ is the number of cells within each chunk and $k$ is the local index ranging from 1 to $N$.

*Step 4*. Re-do the above procedures to the next azimuthal ring until ring average is not needed.

*For variables using the PPM reconstruction*

The procedures in PPM are the same with PLM except for *Step 3*.

*Step 3*. Reconstruct the variables parabolically in each chunk using the following function:

$$F_k = \frac{A}{3N^2}(3k^2 - 3k + 1) + \frac{B}{2N}(2k - 1) + C, \tag{17}$$

where $A$, $B$, and $C$ are constants representing the parabolic function which connects $F_L$ and $F_R$:

$$\begin{aligned}
A &= 3(F_L - F_R - 2F_A) \\
B &= 2(3F_A - 2F_L - F_R) \\
C &= F_L.
\end{aligned} \tag{18}$$

*For vector variables using the PLM reconstruction and Fourier reduction*

The "*Step 0*" in Figure 3 corresponds to a Fourier expansion (reduction) step that is required for vector GCM variables in spherical coordinates before applying the Ring Average process. The main purpose of the Fourier reduction step is to maintain the direction of vectors after Ring Average, especially for the neutral meridional and zonal wind across the pole. Thus only the second and higher Fourier components of





the data in the azimuthal cell is smoothed using the Ring Average filter, while the zeroth and first Fourier

components are kept unchanged. Here are the details of the Fourier expansion process:

*Step 0.* Calculate the Fourier components of the azimuthal data:

$$P_i = A_0 + A_1 \cos\left(2\pi i/N_{total}\right) + B_1 \sin\left(2\pi i/N_{total}\right) + F_i, \tag{19}$$

where $i$ is the thin cell index along the azimuthal direction ranging from $1$ to $N_{total}$, where $N_{total}$ is the total

thin cell number in the azimuthal direction, $F_i$ is the second and higher Fourier components which will be

later reconstructed, $A_0$, $A_1$, $B_1$ are the zero and first Fourier coefficients:

$$
\begin{aligned}
A_0 &= \frac{1}{L}\sum_{i=1}^{i=L} P_i \\
A_1 &= \frac{1}{L}\sum_{i=1}^{i=L} P_i \cos\left(2\pi i/L\right) \\
B_1 &= \frac{2}{L}\sum_{i=1}^{i=L} P_i \sin\left(2\pi i/L\right).
\end{aligned}
\tag{20}
$$

The higher Fourier components $F_i$ are pulled into chunks for the Ring Average processes.

*Step 1-4*: Same with the above PLM methods, except that the reconstructed data $F_i^{'}$ is brought back

together with the first two Fourier components after the reconstruction:

$$P_i = A_0 + A_1 \cos\left(2\pi i/L\right) + B_1 \sin\left(2\pi i/L\right) + F_i^{'}. \tag{21}$$

## 2.2 Ring Average for the Advection Equation

In this section, in order to illustrate the implementation of the Ring Average algorithm in a finite dif-

ference code, we solve the two-dimensional (2D) linear advection equation in the polar geometry as an

example. The code used in the 2D linear advection solver is a main subroutine used in the Ring Average

module for the GCMs. This two-dimensional advection test in polar geometry is also useful to demonstrate

the effectiveness of the finite-difference Ring Average technique in handling a strong, narrow shear flow

near the pole. A fourth-order central finite difference scheme is used to solve the following mass continuity





equation under the incompressible assumption:

$$\frac{\partial \rho}{\partial t} + \mathbf{u} \cdot \nabla \rho = 0, \tag{22}$$

where $\rho$ is the density, and $\mathbf{u}$ is the time-stationary flow velocity defined in polar coordinate $(r, \theta)$. The polar geometry of this test is defined with a resolution of $0.625°$ in both longitude and latitude, with 144 cells in the $r$-direction uniformly distributed between $(0, 1)$ and 576 cells in the $\theta$-direction uniformly distributed between $(0, 2\pi)$.

Figure 4a shows the initial state ($t = 0$) of $\rho$, ranging linearly along the $y$ direction from a magnitude of 2 at the topside to 0.01 at the bottomside:

$$\rho = \begin{cases} 2 & y \geqslant 0.65 \\ (y - 0.65)/(0.65 + 1) * (2 - 0.01) + 2 & y < 0.65. \end{cases} \tag{23}$$

The time-independent flow velocity $\mathbf{u}$ is set as a Gaussian-distributed shear flow towards $-y$ and centered at $x = 0.15$ with a peak velocity of $-1$ and a half width of $0.01$:

$$u = -\exp\left[-\frac{(x - 0.15)^2}{0.01}\right]. \tag{24}$$

As simulation time evolves, a large density gradient occurs near the pole driven by the time-stationary shear flow, with its pattern following the analytical distribution of the flow velocity $\mathbf{u}$. Figures 4a-4c show three snapshots of the density in the linear advection experiment at $t = 0$, $t = 0.75$, and $t = 1.5$ using the finite difference version of the Ring Average technique with PPM reconstructions, as described in Section 2.1. For comparisons, Figures 4d-4f show the corresponding snapshots derived from another simulation using an FFT filter, and Figures 4g-4i show the results at the same simulation time calculated from the fourth-order finite difference scheme without applying any filtering technique. In the simulation using Ring Average, the number of the averaging chunks in each azimuthal ring near the pole is set to be [18 18 18 18 36 36 36 36 72 72 72 72 144 144 144 144], from the first ring to the 16th ring, as indicated by the white circles in the top panels around $80°$N. For the FFT filter, a Fourier expansion is applied in the azimuthal direction at each time step to the fluid density. Waves with frequencies that are higher than the cutoff frequencies are eliminated from the Fourier spectra of prognostic variables. The values of prognostic variables are then





reconstructed through an inverse Fourier transform using the modified Fourier spectra. Each latitude grid

has its own cutoff frequency and the wave number to be cut off in this experiment near the pole is set to be

[1 1 2 2 2 2 4 4 4 4 8 8 8 8 10 10], which is similar to the TIE-GCM FFT filter spectrum [Wang, 1998].

As shown in Figures 4b-4c, density structures with large spatial gradient flow across the pole as time

progresses. Compared with the non-filter case in Figures 4h-4i, no evident numerical instability or artifi-

cial structure occurred when applying the Ring Average technique. In contrast, the density structure using

an FFT filter in Figures 4d-4f exhibits numerical oscillations in the radial direction, together with an ar-

tificial depletion of density near the pole. This density depletion is due to the non-conservative nature of

the FFT method by truncating high spatial frequency wave modes in a linear way. Figures 4j-4l show a

one-dimensional comparison of the density profiles along $x = 0.15$, with the region of averaging chunks

denoted by yellow. The comparisons suggest that the density flow is not noticeably affected by the imple-

mentation of the Ring Average technique in the finite difference solver. Note that the time step used after

applying the Ring Average technique is 0.0001 $s$, which is 25 times larger than that used in the simulation

without Ring Average ($dt = 0.000004s$). Although the FFT filter can result in non-oscillatory solutions in

the finte-difference solver, however, as shown by the one-dimensional profiles in Figure 4l, evident density

oscillations occur near the pole due to numerical instability caused by the FFT method. The cut-off fre-

quency of FFT filter is case-dependent and have a problem of mass-loss, as compared to the Ring Average

method. The advection experiment illustrates that the Ring Average technique is capable of relaxing the

severe time step constraint and resolving large density gradient when passing through the clustered grid

cells near the pole.



**Fig 4** The distribution of density at three simulation snapshots (t=0, 0.75 and 1.5). The first three panels from top show the results with the Ring Average technique, with a FFT filter, and without any filter. The latitude boundaries of Ring Average and FFT filter are marked by white circles in the upper two panels. The bottom panels show the comparison of distributions of density along the line $x = 0.15$ (star) with Ring Average technique, (blue dot) with FFT filter, and (red circle) without filter at the three snapshots, respectively. The number of averaging chunks for the Ring Average technique in each azimuthal ring near the pole is set to be [18 18 18 18 36 36 36 36 72 72 72 72 144 144 144 144].





### 2.3 Ring Average for GCMs

We use the NCAR-TIE-GCM to demonstrate the effectiveness of the Ring Average technique in resolving mesoscale upper atmospheric structures. TIE-GCM is a physics-based 3-D global model that solves the coupled equations of momentum, energy, and continuity for neutral and ion species of the upper atmospheric I-T system, using a fourth-order and centered finite difference scheme to evolve the advection terms on each pressure surface with a staggered vertical grid [Qian et al., 2014; Richmond et al., 1992; Roble et al., 1988]. The TIE-GCM utilizes a spherical coordinate system fixed with respect to the rotating Earth, with geographic latitude and longitude as the horizontal coordinates and pressure surface as the vertical coordinate. Following is a brief introduction of the basic equations in the TIE-GCM.

The thermospheric energy equation is

$$\frac{\partial T_n}{\partial t} = -\mathbf{V}\cdot\nabla T_n + \frac{ge^Z}{p_0 C_p}\frac{\partial}{\partial Z}\left\{\frac{K_T}{H}\frac{\partial T_n}{\partial Z} + K_E H^2 C_P \rho\left[\frac{g}{C_P} + \frac{1}{H}\frac{\partial T_n}{\partial Z}\right]\right\} - w\left(\frac{\partial T_n}{\partial Z} + \frac{R^*T}{C_p\overline{m}}\right) + \frac{Q-L}{C_P}, \quad (25)$$

with temperature $T_n$, time $t$, the vertical coordinate $Z = ln(p_0/p)$, the pressure $p$ and $p_0$ the reference pressure. $g$ is gravity, $K_T$ is the molecular thermal conductivity, $C_P$ is the specific heat per unit mass, $H$ is the pressure scale height, $K_E$ is the eddy diffusion coefficient, $\rho$ is the atmospheric mass density, $\mathbf{V}$ is the horizontal neutral velocity with the zonal and meridional components $u_n$ and $v_n$, $w$ is the vertical velocity defined by $w = dZ/dt$, $R^*$ is the universal gas constant, $\overline{m}$ is the mean atmospheric mass, and $Q$ and $L$ are the heating and cooling rates. The mean molecular mass $\overline{m}$ is determined by

$$\overline{m} = \left[\frac{\Psi_{O_2}}{m_{O_2}} + \frac{\Psi_O}{m_O} + \frac{\Psi_{N_2}}{m_{N_2}}\right], \quad (26)$$

where $\Psi$ and $m$ represent the mass mixing ratio and the molecular mass for the three thermospheric major species $O_2$, $O$, and $N_2$, respectively.

The zonal momentum equation is expressed as

$$\frac{\partial u_n}{\partial t} = -\mathbf{V}\cdot\nabla u_n + \frac{ge^Z}{p_0}\frac{\partial}{\partial Z}\left(\frac{\mu}{H}\frac{\partial u_n}{\partial Z}\right) + \left(f + \frac{u_n}{R_E}\tan\lambda\right)v_n + \lambda_{xx}(u_i - u_n)$$
$$+ \lambda_{xy}(v_i - v_n) - w\frac{\partial u_n}{\partial Z} - \frac{1}{R_E\cos\lambda}\frac{\partial\phi}{\partial\varphi}, \quad (27)$$





and the meridional momentum equation is

$$\frac{\partial v_n}{\partial t} = -\mathbf{V} \cdot \nabla v_n + \frac{g e^Z}{P_0} \frac{\partial}{\partial Z}(\frac{\mu}{H} \frac{\partial v_n}{\partial Z}) - (f + \frac{u_n}{R_E} \tan \lambda) u_n + \lambda_{yy}(v_i - v_n) \\ + \lambda_{yx}(u_i - u_n) - w \frac{\partial v_n}{\partial Z} - \frac{1}{R_E} \frac{\partial \phi}{\partial \lambda}, \tag{28}$$

where $\lambda$ and $\varphi$ represent the geographic latitude and longitude, respectively. $R_E$ is the radius of the Earth,

$\mu$ is the viscosity coefficient which is the sum of eddy and molecular viscosity coefficients, $f$ is the Coriolis

parameter, $\phi$ is the geopotential, $H$ is the pressure scale height, $v_i$ and $u_i$ are the meridional and zonal

$\mathbf{E} \times \mathbf{B}$ ion drift velocities, and $\lambda_{xx}, \lambda_{xy}, \lambda_{yx}, \lambda_{yy}$ are the ion-drag tensor coefficients. The TIE-GCM "vertical

velocity" $w = dZ/dt$ is determined by solving the continuity equation:

$$\frac{1}{r \cos \lambda} \frac{\partial}{\partial \lambda}(v_n \cos \varphi) + \frac{1}{r \cos \lambda} \frac{\partial u_n}{\partial \varphi} + e^Z \frac{\partial}{\partial Z}(e^{-Z} w) = 0. \tag{29}$$

The real vertical velocity is obtained by first integrating the continuing equation (29) over $Z$ to get $w$, and

then multiplying $w$ by the neutral pressure scale height to get the right unit.

The thermospheric major species in the TIE-GCM includes $O_2$, $O$, and $N_2$. The continuity equation for

the mass mixing ratio of $O_2$ and $O$ is given by

$$\frac{\partial \tilde{\Psi}}{\partial t} = -\mathbf{V} \cdot \nabla \tilde{\Psi} - \frac{e^Z}{\tau} \frac{\partial}{\partial Z}[\frac{\overline{m}}{m_{N_2}}(\frac{T_{00}}{T_n})^{0.25} \tilde{\alpha}^{-1} L \tilde{\Psi}] + e^Z \frac{\partial}{\partial Z}[K(z) e^{-Z} \frac{\partial}{\partial Z}(1 + \frac{1}{\overline{m}} \frac{\partial \overline{m}}{\partial Z}) \tilde{\Psi}] \\ + \tilde{S} - \tilde{R} - w \frac{\partial \tilde{\Psi}}{\partial Z}, \tag{30}$$

where $\tilde{\Psi} = (\Psi_{O_2}, \Psi_O)$, $\tau$ is the diffusion time scale and equals to $1.86 \times 10^3 \, s$, $m_{N_2}$ is the molecular mass

for molecular nitrogen, $T_{00} = 273 \, K$ is the standard temperature, $\tilde{\alpha}$ is the matrix operator of the diffusion

coefficients, $K(Z)$ is the eddy diffusion coefficient, and $\tilde{S}$ and $\tilde{R}$ are the production and loss term for these

two species. The diagonal matrix operator $L$ has elements of the form

$$L_{ii} = \frac{\partial}{\partial Z} - (1 - \frac{m_i}{\overline{m}} - \frac{1}{\overline{m}} \frac{\partial \overline{m}}{\partial Z}), \tag{31}$$

where $i = 1, 2$ denote $O_2$ and $O$, respectively. The $N_2$ mass mixing ratio is determined by

$$\Psi_{N_2} = 1 - \Psi_{O_2} - \Psi_O. \tag{32}$$





The minor species in the TIE-GCM are $N(^4S)$, $N(^2D)$, and $NO$. The time scale of $N(^4S)$ is relatively short and thus is considered to be photochemical equilibrium. $N(^4S)$ and $NO$ have longer life times so the transport effects must be taken into account. The governing equation for these two species is

$$\frac{\partial \tilde{\Psi}}{\partial t} = -\mathbf{V} \cdot \nabla \tilde{\Psi} - e^Z \frac{\partial}{\partial Z} \tilde{A}(\frac{\partial}{\partial Z} - \tilde{E}) \tilde{\Psi} + e^Z \frac{\partial}{\partial Z} e^{-Z} K_E(Z)(\frac{\partial}{\partial Z} + \frac{1}{\overline{m}} \frac{\partial \overline{m}}{\partial Z}) \tilde{\Psi} - w \frac{\partial \tilde{\Psi}}{\partial Z} + \tilde{S} - \tilde{R}, \quad (33)$$

where

$$E = (1 - \frac{\tilde{m}}{\overline{m}} - \frac{1}{\overline{m}} \frac{\partial \overline{m}}{\partial Z}) - \tilde{\alpha} \frac{1}{T_n} \frac{\partial T_n}{\partial Z} + \tilde{F} \tilde{\Psi}, \quad (34)$$

where $\tilde{\Psi} = (\Psi_{NO}, \Psi_{N(^4S)})$, $\tilde{A}$ is the vertical molecular diffusion coefficient, $\tilde{S}$ and $\tilde{R}$ are the production and loss terms for each species. Terms in $\tilde{E}$ represent the effects of gravity, thermal diffusion and the frictional interaction with the major species on the vertical profiles of these two species. $\tilde{F}$ is a matrix operator for the frictional interactions, $\tilde{\alpha}$ is the thermal diffusion coefficient, and $\tilde{m}$ is the molecular mass for the two minor species.

The ions of the ionosphere in the TIE-GCM include $O^+$, $O_2^+$, $NO^+$, $N^+$, and $N_2^+$, and the electron density is calculated by chemical equilibrium of these ions. All major ionospheric ions except $O^+$ are assumed as photochemical equilibrium, thus their densities can be calculated simply by balancing the loss and production rates. The $O^+$ density is determined not only by $O^+$ loss and production but also by transportation due to $\mathbf{E} \times \mathbf{B}$ drifts, neutral winds, and field-aligned ambipolar diffusion. The $O^+$ continuity equation can be expressed as

$$\frac{\partial n}{\partial t} = -\nabla \cdot (n\mathbf{V}) + Q - Ln, \quad (35)$$

where $n$ is the $O^+$ density, $Q$ is the production rate, $L$ is the loss rate, and $\nabla \cdot (n\mathbf{V})$ is the transport term. The ion velocity is given by

$$\mathbf{V} = \mathbf{V}_{\parallel} + \mathbf{V}_{\perp} \quad (36)$$

$$\mathbf{V}_{\parallel} = \left\{ \mathbf{b} \cdot \frac{1}{\nu} \left[ \mathbf{g} - \frac{1}{\rho_i} \nabla(P_i + P_e) \right] + \mathbf{b} \cdot \mathbf{U} \right\} \mathbf{b} \quad (37)$$

$$\mathbf{V}_{\perp} = \frac{\mathbf{E} \times \mathbf{B}}{|\mathbf{B}|^2}, \quad (38)$$

where $\mathbf{V}_{\parallel}$ and $\mathbf{V}_{\perp}$ are the parallel and perpendicular velocities with respect to the magnetic field line caused by ambipolar diffusion and neutral winds, and $\mathbf{E} \times \mathbf{B}$ drifts, respectively, $\mathbf{b}$ is a unit vector along the magnetic





field, $\nu$ is the ion-neutral collision frequency, $\mathbf{g}$ is the acceleration due to gravity, $\rho_i$ is the ion mass density,

$P_i$ and $P_e$ are the ion and electron pressures, respectively, $\mathbf{U}$ is the neutral velocity, $\mathbf{B}$ is the magnetic field,

and $\mathbf{E}$ is the electric field.

By assuming a thermal quasi-steady state, the electron energy equation is

$$\sin^2 I \frac{\partial}{H \partial Z}(K^e \frac{\partial T_e}{H \partial Z}) + \sum Q_e - \sum L_e = 0, \tag{39}$$

with $I$ the geomagnetic dip angle, $K^e$ the electron thermal conductivity parallel to the magnetic field, $\sum Q_e$

the sum of all local electron heating rates, and $\sum L_e$ the sum of all local cooling rates.

For the electrodynamics, i.e. the "neutral wind dynamo process", TIE-GCM assumes a steady state

electrodynamics with a divergence free current density $\mathbf{J}$ for longer time scales:

$$\nabla \cdot \left[ \sigma_P(\mathbf{E} + \mathbf{U} \times \mathbf{B}) + \sigma_H \mathbf{b} \times (\mathbf{E} + \mathbf{U} \times \mathbf{B}) + \mathbf{J}_{||} + \mathbf{J_M} \right] = 0, \tag{40}$$

where $\sigma_P$ and $\sigma_H$ are the Pederson and Hall conductivities, and $\mathbf{U}$ is the neutral wind. $\mathbf{J}_{||}$ and $\mathbf{J_M}$ are

the ohmic component of current density parallel to the magnetic field and the non-ohmic magnetospheric

component, respectively.

The ionospheric convection pattern for computing the plasma advection velocity $\mathbf{V}_\perp$ at high latitudes

is specified by either the Heelis et al. [1982] or the Weimer [2005] empirical model, while at the bottom

boundary the migrating tides are specified using the Global-Scale Wave Model [Hagan and Forbes, 2002,

2003]. The current standard version of TIE-GCM (TIE-GCM v2.0) provides two spatial resolution options:

(1) $5° \times 5°$ in horizontal geographic latitude-longitude grid and 1/2 scale height in the vertical direction, and

(2) $2.5° \times 2.5°$ in horizontal geographic latitude-longitude grid and 1/4 scale height in the vertical direction.

In this study, the Ring Average technique is implemented in the TIE-GCM v2.0 to solve the issue of

clustering grid cells near the poles in the development of a high-resolution version of the TIE-GCM. This

technique is applied as a post-processing treatment of the fluid variables including oxygen ion density $O^+$,

neutral temperature $T_n$, thermospheric compositions $\Psi$, meridional, zonal, and vertical winds $(U_n, V_n, w)$

at each time step, with different reconstruction methods (PPM or PLM) for different variables (Table 1).

Due to the use of mpi parallelization in the TIE-GCM in supercomputers, the Ring Average technique

firstly collects the azimuthal data in the root thread, conducts the averaging-reconstruction process and

finally redistributes data into each mpi thread. Figure 5 illustrates Ring Average filters used in the main





## TIEGCM RING-AVERAGE FILTER ALGORITHM

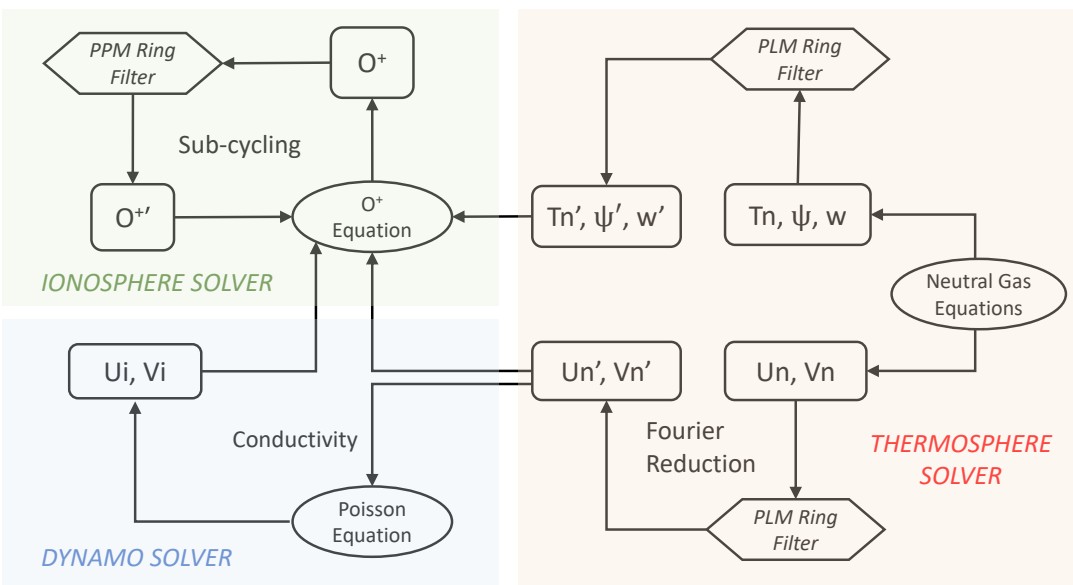

**Fig 5** The main Ring Average Algorithm in the TIE-GCM

algorithms of the TIE-GCM, including the thermosphere solvers in Equations (25-34), the ionosphere solver

for $O^+$ in Equations (35-39), and the dynamo solver for electrodynamic coupling in the Equation (40). For

neutral variables in the thermosphere solver, the Ring Average technique with the PLM reconstruction

method is utilized. Specifically, for the meridional and zonal neutral winds, the second and higher Fourier

components are processed with the PLM Ring Average filter to maintain the direction of vectors across the

pole, as displayed in Figure 5. The oxygen ion ($O^+$) in the ionosphere usually has much sharper gradients

than the neutral variables, e.g., Tongue of Ionization (TOI) structures, thus the PPM method is used in the

reconstruction process to provide high-order accuracy and handle the larger local gradient. Meanwhile, to

balance the numerical stability and computational speed, a sub-cycling technique, which has a smaller time

step for $O^+$ than neutral variables, has been applied in the $O^+$ solver, because the ions can move much faster

than the neutrals with the ExB drifts especially during major geomagnetic storms.

On the basis of the Ring Average technique, a new high-resolution version of TIE-GCM with a hori-

zontal longitude-latitude resolution as high as $0.625° \times 0.625°$ is developed. Table 2 lists the Ring Average

setup used in different TIE-GCM resolutions. The third column in Table 2 represents the number of av-

eraging chunks in each azimuthal ring near the pole, from the first innermost to the outermost rings. For



**Table 1** The Basic Ring Average Settings of Variables (Column 1), and the corresponding Reconstruction Method (Column 2), Fourier Reduction (Column 3), and Sub-cycling (Column 4) in the TIE-GCM.

| Variables | Reconstruction Method | Fourier Reduction | Sub-Cycling |
|---|---|---|---|
| $\Psi_O, \Psi_{O_2}, \Psi_{NO}, \Psi_{N(^4S)}, w, T_n$ | PLM | No | No |
| $U_n, V_n$ | PLM | Yes | No |
| $O^+$ | PPM | No | Yes |

**Table 2** The Ring Average Setup for Different TIE-GCM Horizontal Resolutions (Column 1), Associated with the Number of Longitude Grids (column 2), and the Number of Averaging Chunks in each azimuthal ring near the pole (column 3).

| Horizontal Resolution | Number of Longitude Grids | Number of Chunks |
|---|---|---|
| $2.5° \times 2.5°$ | 144 | [9,18,36,36,72,72,72,72] |
| $1.25° \times 1.25°$ | 288 | [9,9,18,18,36,36,36,36,72,72,72,72,144,144,144,144, 144,144,144,144] |
| $0.625° \times 0.625°$ | 576 | [9,9,9,9,18,18,18,18,36,36,36,36,36,36,36,36,72,72,72, 72,72,72,72,72,144,144,144,144,144,144,144,144,288, 288,288,288,288,288,288,288] |

example, in the first azimuthal ring near the pole of the $0.625° \times 0.625°$ grid resolution, 64 longitude grids

($576/9 = 64$) and 40 longitude degrees ($360°/9 = 40°$) are grouped into a chunk. While in the outermost

filtered ring (around $71.25°$ latitude), one averaging chunk only contains two longitude grids. Table 3 sum-

marizes the information of different spatial resolutions of the TIE-GCM, including the current version of

$2.5° \times 2.5°$ TIE-GCM with the default FFT filter, and $2.5° \times 2.5°$, $1.25° \times 1.25°$, and $0.625° \times 0.625°$ resolu-

tion TIE-GCM with the Ring Average filter, respectively. As the resolution doubles, the time step decreases

approximately linearly rather than quadratically. In practice, the $0.625° \times 0.625°$ resolution of the code runs

about two times faster than real time with 256 processors on the NCAR/CISL Cheyenne supercomputer

system (12 hours for one-day geomagnetic storm simulation), which is at fairly low computational cost for

mesoscale-resolving global simulations. The preliminary results of the high-resolution TIE-GCM will be

shown in the following section.

## 395   3 Applications

To show the capability of the new high-resolution TIE-GCM based on the Ring Average technique in

resolving mesoscale I-T structures, we have simulated the ionospheric and thermospheric variations during

the March 17, 2013 mojor geomagnetic storm as an example. Figure 6 displays the comparison of polar

maps of electron densities between different filter techniques with the $2.5° \times 2.5°$ horizontal resolution.

The electron density is plotted on pressure surface 2, which is near the $F_2$ region peak ($\sim 300$ km altitude).



**Table 3** Comparisons of Horizontal Resolution in Geographic Latitude-Longitude Grid (Column 1), Vertical Resolution (Column 2), Time Step (Column 3), $O^+$ Sub-cycling Time Step (Column 4), and Polar Filter (Column 5) between Different TIE-GCM Versions[a].

| Horizontal Resolution | Vertical Resolution (Scale Height) | Time Step (s) | $O^+$ Sub-cycling Time Step (s) | Polar Filter |
|---|---|---|---|---|
| $2.5° \times 2.5°$ | 1/4 | 60 | - | FFT |
| $2.5° \times 2.5°$ | 1/4 | 60 | 5 | Ring Average |
| $1.25° \times 1.25°$ | 1/4 | 20 | 2 | Ring Average |
| $0.625° \times 0.625°$ | 1/4 | 10 | 0.1 | Ring Average |

[a]In colunms 3-4, the time step corresponds to the cases of geomagnetic storms. The time step and sub-cycling time step would be more relaxed when the geomagnetic activity is quiet.

Figures 6a corresponds to the standard TIE-GCM with the FFT filter, while Figures 6b is the result using the Ring Average technique. Generally, the electron densities in two simulations in Figures 6a-6b are similar below 60°N, with an evident electron density enhancement seen in the afternoon sector and negative storm effects in the morning at 10:50 UT during the storm. The dense ionospheric plasma in the afternoon sector is transported in the anti-sunward direction into the polar cap region by the dusk cell of the convection pattern. Consequently, prominent polar tongue of ionization (TOI) features can be seen as a narrow density plume on the dayside, which stretches from 65°N at noon to latitudes greater than 80°N inside the polar cap. Those TOI features agree well with the polar Global Position System (GPS) total electron content (TEC) observations [e.g., Foster et al., 2005; Thomas et al., 2013]. It is evident that, in Figure 6a, the TOI cannot go through the polar cap region and generates an artificial "hole" structure at above 80°N. This non-physical depletion is associated with the loss of electron density induced by the removal of high frequency in the FFT filter, as also indicated in the advection experiment in Figure 4f. Consequently, the plasma within the TOI accumulates around the "hole" and a "ring-like" structure appears at about 70°N. In contrast, for the Ring Average technique, the electron densities in Figure 6b can successfully flow through the polar cap and arrive at the nightside, which is consistent with Figure 4c. Thus using the Ring Average, the artificial structures no longer exist in the polar cap region, indicating the advantage of Ring Average technique in handling the numerical instability by causing less artificial structures and preserving the real mesoscale structures.

Figure 7 shows the comparison of the polar maps of electron densities between simulations with different spatial resolutions using the Ring Average technique bolstered TIE-GCM. The simulation results after using the Ring Average technique are generally similar among different simulations, with finer structures in higher spatial resolutions. Besides the ionospheric parameters, we have also tested the performance of Ring Average in the thermospheric simulations (not shown here), which indicates that the thermospheric




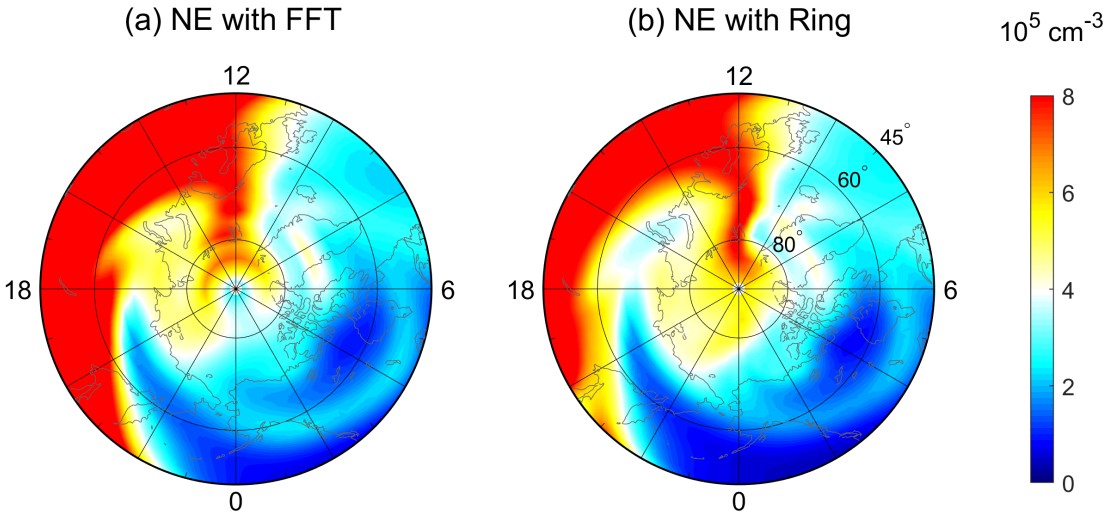

**Fig 6** The simulated polar maps of electron densities using (left) FFT filter and (right) Ring Average at pressure surface 2 (near $F_2$-region peak, $\sim 300$ km) at 10:50 UT on March 17, 2013 as a function of geographic latitude and local time. Both simulations have a horizontal resolution of $2.5° \times 2.5°$. The outer boundary is 45°N geographic latitude.

variables generally converge between different spatial resolutions. The thermospheric temperature, $O/N_2$,

and thermospheric density simulated by two kinds of filters do not show distinct differences as compared

with the ionospheric simulations, due to the relatively smoother variations of neutral parameters. Only

slight deviation exists locally on a smaller scale in the polar thermosphere. The results from Figures 6

and 7 demonstrate that the Ring Average technique can be applied in the finite difference method, which

is usually considered to be less stable than the finite volume scheme. The Ring Average method can suc-

cessfully maintain the numerical stability, even with the structures of large spatial gradients, and conserve

true mesoscale structures. Meanwhile, the Ring Average technique shows advantages of inexpensive com-

putational cost and easy implementation, as indicated by Table 3. By using the Ring Average, the time step

has been greatly relaxed in the ideal advection experiment and the high-resolution TIE-GCM, which would

maintain the computational cost to an acceptable level. Furthermore, the Ring Averaging can be applied

as a post filter after each simulation step and would not require a modification of the underlying code and

make the technique easily applied.

Benefiting from the Ring Average technique, the newly developed high-resolution TIE-GCM has been

applied to explore the mesoscale variations in the I-T system during space weather events. For instance,



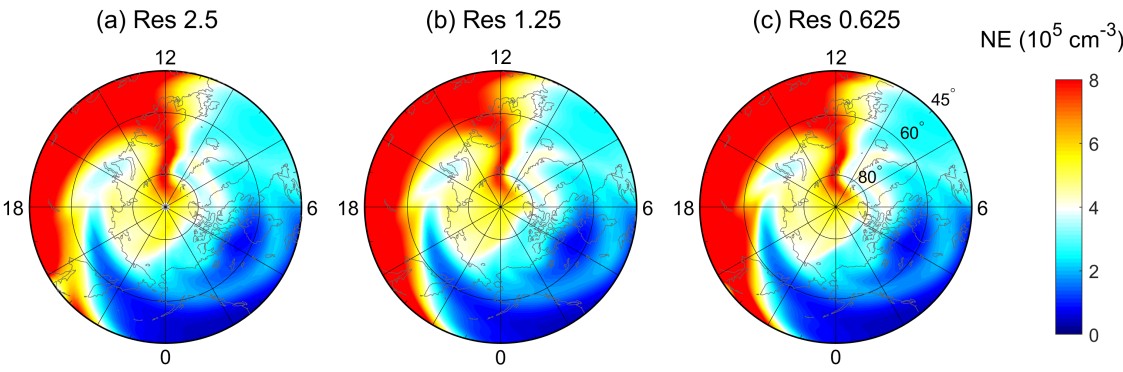

**Fig 7** The polar maps of electron densities at pressure surface 2 (near the $F_2$-region peak, $\sim 300$ km) at 10:50 UT on March 17, 2013 as a function of geographic latitude and local time for (a) $2.5° \times 2.5°$, (b) $1.25° \times 1.25°$, and (c) $0.625° \times 0.625°$ TIE-GCM horizontal resolutions using the Ring Average technique.

based on the $0.625° \times 0.625°$ high-resolution TIE-GCM simulations as well as satellite observations, Dang et al. [2019] have reported the occurrence of double TOIs and carried out a comprehensive study on the dynamic evolution and formation mechanism of double TOIs. Lu et al. [2020] used the high-resolution model to study the ionospheric disturbances such as traveling ionospheric disturbances and storm enhanced density during geomagnetic disturbances. Besides, the high-resolution TIE-GCM has also been utilized to simulate the sub auroral polarization stream [Lin et al., 2019], neutral wind variabilities [Wu et al., 2019], and the responses of the I-T system to solar eclipses [Dang et al., 2018a,b; Lei et al., 2018; Wang et al., 2019]. These works highlight the enhanced capability of high-resolution TIE-GCM in resolving the ionospheric and thermospheric mesoscale structures that is enabled by the Ring Average technique.

Simulating the mesoscale structures also requires a more realistic input from the upper and bottom boundaries of the I-T system, corresponding to the electric field and auroral precipitation from the magnetosphere and the upward propagation of tides and waves from the lower atmosphere, respectively. In the TIE-GCM, these inputs are directly adopted from two empirical models, the Weimer model and GSWM model, which might not necessarily represent the complexity of the actual physical processes from the boundaries. To obtain a more physical upper boundary condition, the CMIT has been developed [Wang et al., 2004; Wiltberger et al., 2004] which couples the LFM global magnetosphere model with the I-T model TIE-GCM. The LFM provides the TIE-GCM with high latitude electric fields and auroral electron precipitations, and the TIE-GCM feeds ionospheric height-integrated conductance back to the LFM. The





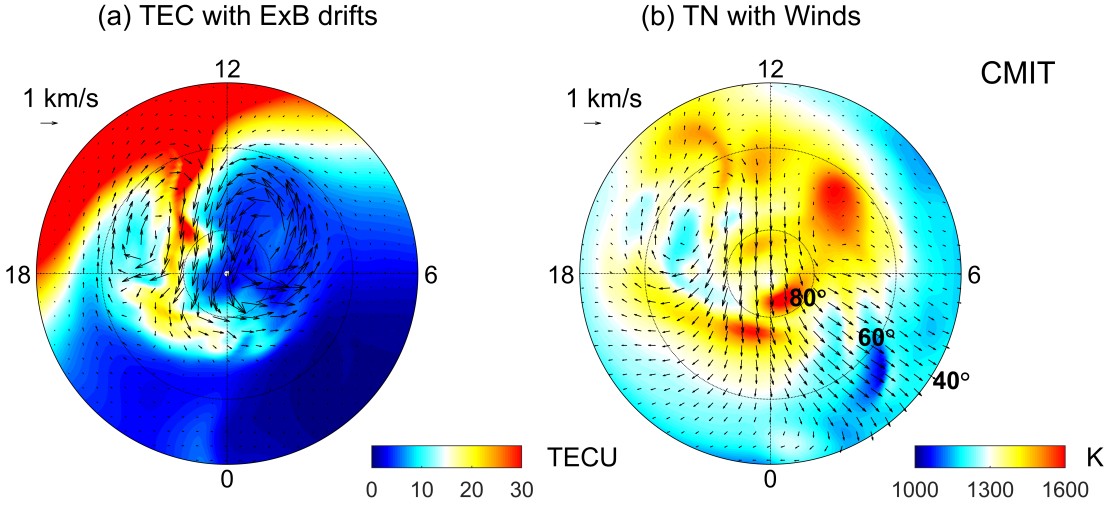

**Fig 8** Polar maps of the (a) total electron content (TEC) and (b) neutral temperature simulated by CMIT as a function of geographic latitude and local time at 17:30 UT on March 17, 2013. The vectors represent the (a) $E \times B$ drifts and (b) horizontal neutral winds, respectively.

standard resolution of the ionosphere and thermosphere in CMIT is $2.5° \times 2.5°$ which is the same as the standard TIE-GCM. By implementing the high-resolution TIE-GCM in CMIT, the thermosphere-ionosphere part in CMIT has a horizontal resolution of $1.25°$ in both latitude and longitude, which is comparable to the magnetospheric resolution of 100 km mapped to the ionospheric reference altitude. Figure 8 shows an example of CMIT simulation of the ionosphere and thermosphere, at 17:30 UT during the March 17, 2013 geomagnetic storm. The TEC in Figure 8a shows more dynamic and finer TOI variations driven by the magnetospheric convection during the storm time. Meanwhile, the thermospheric temperature in Figure 8b also exhibits distinct mesoscale structures, associated with changes in the neutral wind circulation and ion collisional heating. The results illustrate that, with the implementation of the Ring Average technique, the high-resolution CMIT show advantages in resolving the dynamic evolution of mesoscale structures in the coupled magnetosphere-ionosphere-thermosphere system.

Furthermore, the Ring Average technique has also been applied in the WACCM-X which can provide a relatively more realistic bottom boundary for the I-T simulation. The WACCM-X is a whole atmosphere chemistry-climate general circulation model, spanning the range of altitude from the Earth's surface to the upper thermosphere to simulate the entire atmosphere and ionosphere [Liu et al., 2018]. The ionosphere and electrodynamo parts in WACCM-X are the same as in the TIE-GCM. The Ring Average scheme has been





successfully implemented in the $O^+$ transport module of the WACCM-X to get a higher spatial resoultion of the ionosphere. Figure 9 shows the simulation results of 2013 March 17, 2013 geomagnetic storm from WACCM-X. For this simulation, the horizontal resolution is $1.25° \times 0.9°$ in longitude and latitude directions, respectively, and the vertical resolution in the upper atmosphere is 1/4 of scale height. Detailed analyses and exploration of the CMIT and WACCM-X results are beyond the scope of this study and will be studied in the future. On-going efforts also include improving the resolution of vertical direction and electrodynamo of the TIE-GCM and applying the Ring Average technique in high-resolution data assimilation and space weather prediction.

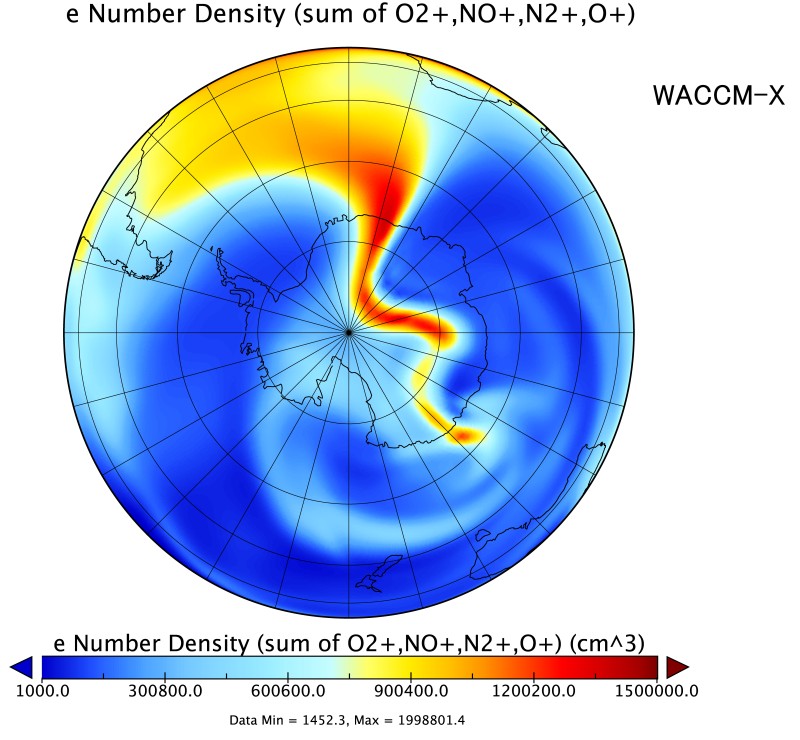

**Fig 9** Polar map of the electron density in the Southern Hemisphere at 14:00 UT on March 17, 2013 from the WACCM-X 1° deg simulation.

## 4 Summary

In summary, a post-processing technique of averaging-reconstruction (Ring Average) algorithm is developed to solve the problem of clustering of azimuthal cells in a spherical coordinate with the finite difference


method. The Ring Average technique is conducted based on a reduced effective polar grid, by first averaging quantities within azimuthal effective "chunks" and then re-constructing them within each chunk. The Ring Average technique shows advantages of inexpensive computational cost, easy implementation, time step relaxation, and maintenance of the mesoscale structures without introducing artifacts, which allows for the development of high resolution GCMs to resolve mesoscale structures. We have developed a new version of the TIE-GCM which has a horizontal resolution of $0.625° \times 0.625°$ in geographic longitude-latitude grid by implementing the Ring Average technique as a post-process step. The non-physical "hole" and "ring" structures, which are induced by FFT filter in the previous TIE-GCM version, no longer exist in the high-resolution TIE-GCM associated with the Ring Average technique. The simulation results illustrate that the high-resolution TIE-GCM is capable of resolving the mesoscale structures in the I-T system during a geomagnetic storm event. Moreover, the Ring Average scheme has also been implemented in CMIT and WACCM-X to enable a high spatial resolution self-consistent simulations of the whole geospace from ground to the magnetosphere.

*Code Availability*   The Ring Average technique and numerial experiments used in this study is available at https://doi.org/10.5281/zenodo.3719295. The default TIE-GCM is developed by NCAR and is available at http://www.hao.ucar.edu/modeling/tgcm/tie.php.

*Author contribution*   TD developed the model code, performed the simulations and wrote the manuscript. BZ, JL, and KS proposed the original idea and revised the manuscript. WW and AB helped to develop the code and edited the manuscript. HL and KP contributed to couple the high-resolution TIEGCM to WACCM-X and CMIT, respectively.

*Competing interests*   The authors declare that they have no conflict of interest.

*Acknowledgments*

This work was supported by the B-type Strategic Priority Program of the Chinese Academy of Sciences (XDB41000000), the National Natural Science Foundation of China (41831070, 41974181), and the Open Research Project of Large Research Infrastructures of CAS - "Study on the interaction between low/mid-latitude atmosphere and ionosphere based on the Chinese Meridian Project". Dang T. was supported by the National Natural Science Foundation of China (41904138), the National Postdoctoral Program for Innovative Talents (BX20180286), the China Postdoctoral Science Foundation (2018M642525) and the Fundamental Research Funds for the Central Universities. HLL's work was supported by the National Center



for Atmospheric Research, which is a major facility sponsored by the National Science Foundation under Cooperative Agreement No. 1852977. The National Center for Atmospheric Research is sponsored by the National Science Foundation. We would like to acknowledge high-performance computing support from Cheyenne (doi:10.5065/D6RX99HX) provided by NCAR's Computational and Information Systems Laboratory, sponsored by the National Science Foundation (NSF).

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
