# Peer review of "Azimuthal Averaging-Reconstructing Filtering Techniques for"

_Geoscientific Model Development, 2020_

## Referee Comment (RC1) · Anonymous Referee #1 · 20 Oct 2020

Summary: The manuscript introduces a new application of the Ring Average technique to be used in Global Circulation Models (GCMs). With this new technique, the numerical constraints imposed on the timestep could be eased, especially at the polar regions where azimuthal grid dimensions become very small. The paper demonstrates the feasibility of the method first in a simpler problem, in which the advection equation is solved on a spherical domain. The results from the Ring Average technique applied solution are compared with the Fast Fourier Transform (FFT) filter applied solution and a 4th order numerical solution. Ring Average method had a better performance re-

quiring a smaller time step, further establishing its feasibility in improving the numerical resolution around the polar region. The second application was conducted by applying FFT Filter to the TIE-GCM results and using the Ring Average technique implemented version of TIE-GCM for the 17 March 2013 storm. The results showed that the Ring Average technique was able to preserve the directions of vector properties and didn't introduce numerical artifacts that are seen in the polar region with the FFT applied simulation results. The Ring Average technique is a non-intrusive method to increase spatial resolution without suffering from small timestep limitations for convergence and stability in GCMs.

General Comments: The manuscript does a commendable job in presenting the new Ring Average method. The need for remedying the resolution problem in GCMs at the polar regions is sufficiently laid out with proper referencing to the literature. The derivation and the implementation of the technique are intuitive and coherent. The results from the introduced model are evaluated in two different examples and clearly demonstrated an advancement in performance with a strong potential to further the science. In addition, the code repositories are provided in the "Code Availability" section and the codes are presented in a user-friendly and self-explanatory manner. Overall, the paper is exceptionally well-written and organized.

Specific Comments: 1. Please consider revising Key Point 2 to clarify what is meant by "more complicated geoscientific models". The paper only discusses the application to TIE-GCM and WACCM-X models and it is not immediately clear how these models compare geoscientifically. 2. Please consider adding "forward" to Line 134 to read as "a central difference forward Euler".

Technical Comments: 1. Please consider replacing "On the other hand" with "In addition" or "Furthermore" on Line 171. 2. Please extend Figure 1 caption to include how the information in Lines 123-124 about the number of "chunks".

---

## Referee Comment (RC2) · Anonymous Referee #2 · 26 Oct 2020

This paper addresses means of overcoming the problems introduced by the small cell size near the poles when solving equations numerically on a spherical grid. It describes a solution using the Ring Average technique which is then illustrated using a widely-used upper atmosphere community model, the TIE-GCM. The authors show that the Ring Average technique allows the TIE-GCM to be run at significantly higher spatial resolution without increasing computational costs significantly or introducing numerical artifacts that other methods do. I found the paper to be clear and well written. I recommend publication after correction of some minor errors and typos.

[Figure]

Minor errors: Line 305: should read "continuity equation" Line 331: replace "transportation" with "transport" Line 349: delete "a" Line 398: "major" is mistyped Line 401: "Figure 6a" and "Figure 6b" – remove the "s" Line 479: "resolution" is mistyped
* * *

---

## Short Comment (SC1) · 27 Oct 2020

Dear authors,

in my role as Executive editor of GMD, I would like to bring to your attention our Editorial version 1.2:

https://www.geosci-model-dev.net/12/2215/2019/

This highlights some requirements of papers published in GMD, which is also available

on the GMD website in the 'Manuscript Types' section: http://www.geoscientific-model-development.net/submission/manuscript_types.html

In particular, please note that for your paper, the following requirements have not been met in the Discussions paper:

- "The main paper must give the model name and version number (or other unique identifier) in the title."

- Code must be published on a persistent public archive with a unique identifier for the exact model version described in the paper or uploaded to the supplement, unless this is impossible for reasons beyond the control of authors. All papers must include a section, at the end of the paper, entitled "Code availability". Here, either instructions for obtaining the code, or the reasons why the code is not available should be clearly stated. It is preferred for the code to be uploaded as a supplement or to be made available at a data repository with an associated DOI (digital object identifier) for the exact model version described in the paper. Alternatively, for established models, there may be an existing means of accessing the code through a particular system. In this case, there must exist a means of permanently accessing the precise model version described in the paper. In some cases, authors may prefer to put models on their own website, or to act as a point of contact for obtaining the code. Given the impermanence of websites and email addresses, this is not encouraged, and authors should consider improving the availability with a more permanent arrangement. Making code available through personal websites or via email contact to the authors is not sufficient. After the paper is accepted the model archive should be updated to include a link to the GMD paper.

Please provide the TIE-GCM version number in the title of your revised manuscript. Additionally, it might be useful to define a version number for the Ring Average technique.

[Figure]

Yours, Astrid Kerkweg

---

## Author Comment (AC3) · 18 Dec 2020

Dear editor,

Thanks very much for your comments. We have used "TIEGCM 2.0r" as the model version and uploaded the source files of the model directed to Github as suggested (https://github.com/dangt-ustc/TIEGCM2.0r). The Ring Average technique and numerical experiments used in this study is available at https://doi.org/10.5281/zenodo.3719295. In addition, to broaden the application of the

ring average method in GCMs with finite difference scheme and spherical geometry, our contributing authors suggested to change the manuscript title as: "Averaging-reconstructing filtering techniques for finite-difference general circulation models in spherical geometry".

---

## Author Response (AR2)

**Responses to the comments of Referee 1**

The manuscript introduces a new application of the Ring Average technique to be used in Global Circulation Models (GCMs). With this new technique, the numerical constraints imposed on the timestep could be eased, especially at the polar regions where azimuthal grid dimensions become very small. The paper demonstrates the feasibility of the method first in a simpler problem, in which the advection equation is solved on a spherical domain. The results from the Ring Average technique applied solution are compared with the Fast Fourier Transform (FFT) filter applied solution and a 4th order numerical solution. Ring Average method had a better performance requiring a smaller time step, further establishing its feasibility in improving the numerical resolution around the polar region. The second application was conducted by applying FFT Filter to the TIE-GCM results and using the Ring Average technique implemented version of TIE-GCM for the 17 March 2013 storm. The results showed that the Ring Average technique was able to preserve the directions of vector properties and didn't introduce numerical artifacts that are seen in the polar region with the FFT applied simulation results. The Ring Average technique is a non-intrusive method to increase spatial resolution without suffering from small timestep limitations for convergence and stability in GCMs.

General Comments: The manuscript does a commendable job in presenting the new Ring Average method. The need for remedying the resolution problem in GCMs at the polar regions is sufficiently laid out with proper referencing to the literature. The derivation and the implementation of the technique are intuitive and coherent. The results from the introduced model are evaluated in two different examples and clearly demonstrated an advancement in performance with a strong potential to further the science. In addition, the code repositories are provided in the "Code Availability" section and the codes are presented in a user-friendly and self-explanatory manner. Overall, the paper is exceptionally well-written and organized.

Thanks very much for your thoughtful and positive comments.

Specific Comments: 1. Please consider revising Key Point 2 to clarify what is meant by "more complicated geoscientific models". The paper only discusses the application to TIE-GCM and WACCM-X models and it is not immediately clear how these models compare geoscientifically.

**Response:** Thanks for your comments. We have revised Key Point 2 as "The Ring Average technique is applied to develop a 0.625×0.625 high-resolution TIE-GCM and more complicated geoscientific models with polar/spherical coordinates and finite difference numerical schemes".

2. Please consider adding "forward" to Line 134 to read as "a central difference forward Euler".

**Response:** Added as suggested in Line 133.

Technical Comments: 1. Please consider replacing "On the other hand" with "In addition" or "Furthermore" on Line 171.

**Response:** We changed it as "Furthermore" in Line 170 as suggested.

2. Please extend Figure 1 caption to include how the information in Lines 123-124 about the number of "chunks".

**Response:** We added "For example, the 144 azimuthal cells in the most inside (highest latitude) grid (Figure 1a) have been grouped to 9 effective cells (chunks), with 16 original cells in each chunk. In the effective grid, the numbers of chunks from inside to outside are 9, 9, 18, 18, 36, 36, 72, 72, 72, and 72, respectively." In the caption of Figure 1.

**Responses to the comments of Referee 2**

This paper addresses means of overcoming the problems introduced by the small cell size near the poles when solving equations numerically on a spherical grid. It describes a solution using the Ring Average technique which is then illustrated using a widely used upper atmosphere community model, the TIE-GCM. The authors show that the Ring Average technique allows the TIE-GCM to be run at significantly higher spatial resolution without increasing computational costs significantly or introducing numerical artifacts that other methods do. I found the paper to be clear and well written. I recommend publication after correction of some minor errors and typos.

Thanks very much for your thoughtful and positive comments.

Minor errors: Line 305: should read "continuity equation" Line 331: replace "transportation" with "transport" Line 349: delete "a" Line 398: "major" is mistyped Line 401: "Figure 6a" and "Figure 6b" – remove the "s" Line 479: "resolution" is mistyped.

**Response:** We corrected the English errors as suggested.

**Reply to the Executive Editor**

This highlights some requirements of papers published in GMD, which is also available on the GMD website in the 'Manuscript Types' section: http://www.geoscientific-modeldevelopment. net/submission/manuscript_types.html

In particular, please note that for your paper, the following requirements have not been met in the Discussions paper: "The main paper must give the model name and version number (or other unique identifier) in the title."

Code must be published on a persistent public archive with a unique identifier for the exact model version described in the paper or uploaded to the supplement, unless this is impossible for reasons beyond the control of authors. All papers must include a section, at the end of the paper, entitled "Code availability". Here, either instructions for obtaining the code, or the reasons why the code is not available should be clearly stated. It is preferred for the code to be uploaded as a supplement or to be made available at a data repository with an associated DOI (digital object identifier) for the exact model version described in the paper. Alternatively, for established models, there may be an existing means of accessing the code through a particular system. In this case, there must exist a means of permanently accessing the precise model version described in the paper. In some cases, authors may prefer to put models on their own website, or to act as a point of contact for obtaining the code. Given the impermanence of websites and email addresses, this is not encouraged, and authors should consider improving the availability with a more permanent arrangement. Making code available through personal websites or via email contact to the authors is not sufficient. After the paper is accepted the model archive should be updated to include a link to the GMD paper.

Please provide the TIE-GCM version number in the title of your revised manuscript. Additionally, it might be useful to define a version number for the Ring Average technique.

**Response:** Thanks very much for your comments. We have used "TIEGCM 2.1" as the model version and uploaded the source files of the model directed to Github as suggested (https://github.com/dangt-ustc/TIEGCM2.1). The Ring Average technique and numerical experiments used in this study is available at https://doi.org/10.5281/zenodo.3719295. In addition, to broaden the application of the ring average method in GCMs with finite difference scheme and spherical geometry, our contributing authors suggested to change the manuscript title as: "Averaging-reconstructing filtering techniques for finite-difference general circulation models in spherical geometry". We added these statements in Code Availability section in Lines 496-498.